# Effect of Dexamethasone and Route of Administration on Sow Farrowing Behaviours, Piglet Delivery and Litter Performance

**DOI:** 10.3390/ani12070847

**Published:** 2022-03-28

**Authors:** Sophia A. Ward, Roy N. Kirkwood, Yunmei Song, Sanjay Garg, Kate J. Plush

**Affiliations:** 1School of Animal and Veterinary Sciences, University of Adelaide, Roseworthy, SA 5371, Australia; roy.kirkwood@adelaide.edu.au; 2Clinical and Health Sciences, University of South Australia, Adelaide, SA 5000, Australia; may.song@unisa.edu.au (Y.S.); sanjay.garg@unisa.edu.au (S.G.); 3Sunpork Group, 1/6 Eagleview Place, Eagle Farm, QLD 4009, Australia; kate.plush@sunporkfarms.com.au

**Keywords:** dexamethasone, farrowing, sow behaviour, piglet performance

## Abstract

**Simple Summary:**

The pain experienced during labor is one that is shared universally. When sows experience the pain of labor for the first time, the levels of discomfort can be so stressful that they lash out aggressively at their piglets. Sows new to the birthing experience may also have problems with delivery or resist nursing the litter for extended periods of time. To help younger sows during and after delivery, we treated a group with dexamethasone, a strong anti-inflammatory treatment. It was predicted that this anti-inflammatory would be able to provide some relief from the inflammatory pain associated with labor and help younger sows with their birthing processes and nursing of their litter. As a hormone that can easily pass through cell walls, it was also predicted that dexamethasone could pass directly through the vaginal membrane of a sow for a non-injectable treatment alternative.

**Abstract:**

The inflammatory pain and stress some crated sows experience during farrowing has attendant risks of piglet-directed aggression, reduced teat exposure and hindered post-partum recovery. To counter this, the steroidal anti-inflammatory compound, dexamethasone, can be administered. To measure the potential for mucosal absorption as an alternative to injection, the permeability of porcine vaginal mucosa to dexamethasone was demonstrated using Franz cell diffusion. These studies found dexamethasone treatment diffused through vaginal mucosa at a constant rate, with 52.37 ± 5.54% permeation in 6 h. To examine in vivo effects on farrowing outcomes, dexamethasone was administered to gilts and parity one sows on the day of expected farrowing. We hypothesized that it would provide relief from farrowing discomfort and reduce behaviours threatening piglet survival. Sows were randomly assigned to receive dexamethasone as an intramuscular injection (*n* = 23); dexamethasone applied topically into the vagina (*n* = 20), or to receive no dexamethasone (*n* = 23). Sows (*n* = 66) and piglets (*n* = 593) were monitored for performance indicators during farrowing and early lactation. A subset of sows (*n* = 24) was also video monitored continuously over 24 h for behaviours associated with pain, postural changes and piglet interactions. No differences were observed between treatment for farrowing performance, piglet survival or behavioural changes for sows experiencing their first or second farrowing (*p* > 0.05), rejecting the hypothesis that corticosteroid administration will improve sow farrowing performance. This investigation did, however, show that dexamethasone can permeate through porcine vaginal mucosa and so can be administered as a non-injectable treatment.

## 1. Introduction

Giving birth can be a stressful and painful event for sows [1], with first-time farrowings being particularly problematic. Primiparous sows tend to be more restless in farrowing crates [2] and susceptible to piglet-directed aggression [1,2,3,4,5]. Sows savaging piglets may not necessarily reflect poor maternal ability, but rather a nervous reaction to the pain of farrowing [5,6] in a restricting, crated environment [3]. Previous reports on sow savaging found a correlation with more restless behaviours leading up to the expulsion of the first piglet [4], which is a reportedly painful stage of parturition [1].

Farrowing behaviour and analgesic use have been studied previously [7,8,9] with minimal effect on sow performance or subsequent piglet survival. Unlike previously used anti-inflammatories, the use of a steroidal anti-inflammatory may be more effective for targeting relief across multiple sites of inflammatory tissue. Dexamethasone is a synthetic glucocorticoid with potent anti-inflammatory properties [10]. The potency and multi-targeted action of this drug may provide greater relief compared to other treatments and reduce aggressive behaviour exhibited by gilts (P0) and first-parity (P1) sows. Previous investigations into the use of anti-inflammatories [7,8,9] report administering treatment at the onset of farrowing or immediately after, once the sow has already experienced the pain of piglet expulsion [1]. If the timing of farrowing was controlled, the anti-inflammatory could be administered in the hours leading up to parturition and may reduce the incidence of stress-induced piglet-directed aggression. Because of dexamethasone’s prolonged biological half-life (36–72 h) [11], the analgesic effects may last beyond parturition and into early lactation. Reducing discomfort may encourage the sow to lie in the same position rather than making many postural changes [7,8,12], increasing teat exposure and reducing the incidence of piglet overlay. A previous investigation into the use of dexamethasone prior to farrowing found a small improvement to piglet daily gain when gilts were treated on the day of an induced farrowing [13]. By observing farrowing and early lactation behaviour, it can be determined what effect, if any, dexamethasone has on the periparturient and early lactation sow behaviours that could subsequently benefit piglet survival and growth.

A concern with using dexamethasone to relieve discomfort is the need for intramuscular injection. Injecting a young sow in the hours leading up to parturition may trigger a stress response, possibly nullifying potential benefits provided by the analgesic. As a steroid hormone, dexamethasone could potentially enter the bloodstream by diffusing through the vaginal mucosa, thus removing the need for injection. A Franz cell diffusion test can be used to measure the permeability of the vaginal mucosa and so assess the potential bioavailability of the drug when administered by this route [14]. It was hypothesised that dexamethasone would cross the vaginal mucosa and have an in vivo effect on sow behaviours and/or piglet neonatal survival.

## 2. Materials and Methods

### 2.1. Permeation of Dexamethasone through Porcine Vaginal Mucosa

To assess vaginal permeability in vitro, simulated vaginal fluid (SPVF) was prepared using the composition reported by Owen and Katz for humans [15] and adjusted with NaOH to pH 7 [16,17] to simulate conditions in the sow vagina [18,19]. Porcine vaginal tissue was obtained from a local abattoir (Murray Bridge, SA, Australia) at slaughter and transported to the laboratory in SPVF on ice. The vaginal mucosa was rinsed three times with saline, stripped from underlying connective tissue and muscle and stored at −20 °C in aluminum foil for future use. When required, the vaginal mucosa was hydrated in SPVF at room temperature and mounted onto a Franz diffusion cell. Then, 2.5 mg (500 µL) of aqueous dexamethasone sodium phosphate treatment (Dexapent™, Troy Laboratories, Glendenning, NSW, Australia) and 200 µL SPVF was added to the mucosal surface in the donor chamber of the Franz cell. Water was heated in a water bath to 37 ± 1 °C and pumped around the receptor chamber. The acceptor solution (filtered SPVF) was maintained at 37 ± 1 °C and mixed with a magnetic stirrer throughout the experiment. Franz cell chambers (PermeGear, Hellertown, PA, USA) were set up similar to the schematic in Figure 1.

At designated time points (0.25, 0.5, 1, 1.5, 2, 3, 4, 5 and 6 h), 100 µL of receptor fluid (SPVF) was sampled from the sampling funnel and replaced with an equal volume of fresh SPVF. All samples were prepared for liquid chromatography by adding 50 µL samples to 50 µL mobile phase and vortexed before HPLC analysis.

The separation system consisted of a Lux Cellulose-1 column (Phenomenex Australia, Lane Cove, NSW, Australia) and 1% formic acid in acetonitrile (solvent A) and 2% formic acid in water (solvent B) in a 50:50 ratio. The method was based on the isocratic method used by Karatt et al. [20] with mobile phase A acidified to obtain sharper peak resolution. Flow rates were set at 0.6 mL/min with the column temperature at 50 °C and detected using a wavelength of 241 nm. Calibration graphs were constructed by plotting the peak area with their corresponding concentrations of dexamethasone (linearity range: 2.5–300 ug/mL). The sum of the two observable peaks not present in blank SVF was calculated with r*^2^* = 0.99 (Figure 2).

Permeability was calculated using the following formula:*x* = ((*TsV*)/*OC*) × 100% 
where

*x* = Cumulative amount of drug through vaginal mucosa (%)

*V* = Total volume in Franz cell (5.0 mL)

*OC* = Amount of drug administered in donor compartment (2.5 mg)

*Ts =* Concentration of the sample taken from Franz cell acceptor solution

Release parameters for the permeation of dexamethasone through porcine vaginal mucosa were calculated using the analysis program created by Zhang et al. [21].

### 2.2. Effects of Dexamethasone on Sow and Piglet Performance

#### 2.2.1. Animal Management

Large White × Landrace gilts and P1 sows (sows) were moved into individual farrowing crates one week before their expected due dates of 116 d after the last insemination. Gilts and sows were fed twice daily with a commercial diet formulated to meet all nutrient requirements and had free access to fresh water. At 114 d of gestation, sows received vulva injections of 125 µg prostaglandin analogue, cloprostenol (Juramate^®^, Jurox Pty, Ltd., Rutherford, NSW, Australia), at 0700 and 1300 h to induce sows to farrow on day 115 of gestation. At 0800 h on day 115, sows were randomly assigned to receive 20 mg dexamethasone (Dexapent™, Troy Laboratories, Glendenning, NSW, Australia) either by intramuscular injection (*n* = 23; DexInj), by topical vaginal mucosal deposition (*n* = 20; DexTop) or to serve as non-treated controls (*n* = 23) with equal parity distribution. To administer the DexTop treatment, a thin sterile tube was inserted 20 cm into the vagina, and treatment was administered followed by a 0.5 mL saline flush.

#### 2.2.2. Data Collection

Farrowing duration, total born litter size, stillbirths, incidence of dystocia and piglet overlay in the first 24 h postpartum were recorded as indicators of sow performance. If the piglet delivery interval exceeded 45 min, obstetric assistance was provided for sows, and it was recorded as a dystocia event. The estimated colostrum intake of piglets was calculated using their birth and 24 h weights and the equation proposed by Devillers et al. [22].
*CI* = −217.4 + 0.217 × *t* + 1861019 × *BW*/*t* + *BWb* × (54.8 − 1861019/*t*) × (0.9985 − 3.7 × 10 − 4 × *tfs* + 6.1 × 10 − 7 × *t*^2^*fs*)
where *CI* = colostrum intake (g), *BWb* = piglet body weight at birth, *BW* = piglet body weight at 24 h and *t* = time elapsed from birth to first suckling (min).

Devillers et al. [22] proposed that the interval of elapsed time from birth to first suckling can be estimated as between 15 and 30 min without major error. In our study, the average interval was 20 min.

#### 2.2.3. Farrowing Behaviour

A subset of sows was video recorded for 24 h from the onset of farrowing using CCTV cameras mounted above each farrowing crate. Sows were assigned either to the DexInj (*n* = 8), DexTop (*n* = 9) or Control (*n* = 9) treatment group. Continuous state and point behaviour observations were made by one observer using the ethogram program BORIS. Potential behavioural indicators of pain were based on Ison et al. [23] (Table 1).

#### 2.2.4. Statistics

Data were analyzed using IBM SPSS v20 statistical software. For the primary outcome, effects of dexamethasone on observational behaviour, data were analyzed using a general linear model with a negative binomial distribution. The binary measurements in this investigation (incidence of stillbirth, dystocia, overlay and piglet-directed aggression) were assessed using a generalized linear model fit with binomial distribution. Other outcomes of interest (farrowing duration, piglet birth interval, total piglets born, total piglets born alive and litter size weaned) were measured with a linear mixed model. All data pertaining to sows was fit with treatment (DexInj, DexTop or Control) and parity (gilt or first-parity sow) as fixed effects. The model included sow ID and room (identical farrowing rooms 4 and 5) as random effects.

For measurements pertaining to the piglet (colostrum intake, survival of piglets to 24 h and piglet survival to weaning), outcomes were assessed using a general linear model with sow as the subject and birth order as the repeated measure. Fixed effects included sow treatment (DexInj, DexTop or Control), piglet gender (male/female) and birth weight group (<1.0 kg, low; 1.1–1.35 kg, medium; >1.35 kg, heavy) All data in the investigation were analysed with a confidence limit set at 95% (*p* < 0.05).

## 3. Results

### 3.1. In Vitro Permeability of Dexamethasone through the Vaginal Mucosa

Over the 6 h of the Franz cell test, dexamethasone passed through the vaginal mucosa in an increasing linear function (Figure 3).

The rate at which dexamethasone diffused across the vaginal mucosal membrane is best described by Makoid–Banakar, with an *R*^2^ = 0.9851 and a magnitude of data or AIC = 33.67 (Figure 4, Appendix A).

### 3.2. Farrowing Performance Parameters

As shown in Table 2, treatment had no significant effect on farrowing performance, with no differences in the duration of farrowing (*p* = 0.214), piglet birth interval (*p* = 0.289) or stillbirths (*p* = 0.655), although gilts had comparatively shorter birth intervals compared to P1 sows (*p* = 0.006; Gilts = 11.53 ± 1.3 min; Sows = 18.06 ± 1.9 min). There was also a trend towards higher incidence of dystocia for P1 sows compared to gilts (*p* = 0.068; Gilts = 28 ± 7%; P1 = 54 ± 12%), but no differences between treatment groups was evident (*p* = 0.263). No treatment effects were observed for the colostrum intake of piglets (*p* = 0.718), but differences in intake were observed across the three piglet birthweight groups, with intake increasing with increasing body weight (BW) (*p* = 0.001; Low BW = 278.1 ± 7.0 g; Medium BW = 326.2 ± 4.9 g; Large BW = 349.4 ± 5.0 g). The provision of dexamethasone had no effect on incidence of overlay (*p* = 0.393) or piglet survival in the first 24 h (*p* = 0.872), although a trend was observed for survival to weaning (*p* = 0.094) (Table 2).

Over the 24 h observational period, no differences were observed in individual or total pain behaviours among treatments (*p* > 0.05; Table 3). The incidence of piglet-directed aggression and time spent on the side were all similar between treatment groups.

## 4. Discussion

### 4.1. Franz Cell Permeation Test

Within 6 h, half of the dexamethasone passed through the sow vaginal mucosa, closely following Makoid–Banakar release model kinetics. As predicted, the lipophilic properties of the steroid enabled rapid diffusion across the vaginal mucosal membrane. This rate of diffusion in the present study was slower than permeability reported by Zang et al. [24], who found 60% of dexamethasone sodium phosphate in a film passed through rabbit buccal mucosa within the first 2.5 h. Differences between release studies could be due to differences in an animal model (sow vs. rabbit), mucosa type (vaginal vs. buccal) and/or properties of drug delivery formulation (injectable solution vs. buccal film). With evidence of permeation, future investigations should track the concentration of dexamethasone in sow plasma over time, measuring the concentration with HPLC–mass spectrometry.

Although the treatment used in our study is a dexamethasone product, the two definite peaks present in chromatographs suggest possible traces of another active constituent within the formulation [25]. When Xiao et al. [25] tested a solution with 1% betamethasone in pure dexamethasone, small and large peaks presented on the chromatographs. These peaks increased with increasing concentrations of the solution, similar to what was found with the chromatograph in our study. Bentamethasone is a chemical isomer with similar anti-inflammatory properties to dexamethasone [25].With this considered, the permeation rate of dexamethasone through the vaginal mucosa cannot be definitively defined without clarifying these two peaks against pure betamethasone standard reference. Our chromatographs did give evidence for the passage of dexamethasone through porcine vaginal mucosa, which concurs with our initial hypothesis.

### 4.2. Animal Treatment

With the confirmation of dexamethasone permeating through porcine vaginal mucosa in vitro, we tested it in vivo by deposition onto sow vaginal mucosa. Our data did not show evidence of dexamethasone affecting sow performance or farrowing behaviours. These results are similar to studies using non-steroidal anti-inflammatory (NSAIDs) meloxicam [8], and analgesic butorphanol [7] on post-partum sows. Mainau et al. [8] proposed that the lack of NSAID effect on farrowing performance was the result of administering treatment too late, but treatment in our study was administered before the expulsive phase of farrowing. Our data would suggest the lack of treatment effect on sow behaviour was not the result of administration before or after farrowing onset. Further, the potency of the anti-inflammatory agent used may also not be a critical factor in changing sow performance, as the use of the potent analgesic, butorphanol showed no differences to pain related behaviours within the first 48 h postpartum [7]. What was observed was a significant reduction in posture changes during the 48 h after treatment, which may have reduced the risk of piglet crushing during the nursing period. This may explain why a trend was observed in our data for improved piglet survival for dexamethasone-treated sows.

Lay et al. [26] suggested that anti-inflammatory compounds have little effect on the farrowing sow due to the restrictive nature of farrowing crates. An increase in the nociceptive threshold is mediated by endogenous opioids, which can be inhibited when a sow is restricted from maternal behaviours leading up to parturition [1]. Nowland et al. [27] observed fewer pain-related behaviours (tail flicking, back leg forward, and straining) when sows were housed in open pens over traditional crates. In another investigation, Nowland et al. [28] found less pain-related behaviours in crated sows when they had access to straw in the lead up to farrowing (*n* = 12) [29]. Although the number of sows used for the behavioral studies was similar to those reported by Nowland et al. [28], a larger sample size of predominately primiparous sows should increase the chances of observing restless behaviours, aggressive tendencies, or animals with a higher susceptibility to pain during parturition. Why the provision of an anti-inflammatory does not provide the same analgesic response is something that should be assessed further, particularly if it is coupled with an improvement in farrowing performance. The use of anti-inflammatories may be more beneficial in sow herds that have higher preweaning mortality rates [29] or pre-existing conditions where analgesia would alleviate discomfort [30,31]. Additionally, the effects of dexamethasone may be more evident in larger populations, as the levels of discomfort experienced during parturition can be subjective to the individual [1].

## 5. Conclusions

Administering dexamethasone on the day of an induced farrowing did not affect sow behaviours during parturition and early lactation. This would imply that some level of pain and/or discomfort is normal during parturition and the immediate post-partum period and, as such, would not be responsive to anti-inflammatory treatments. It is possible that a beneficial effect of steroid would be evident only under conditions of abnormal levels of pain or distress, as potentially indicated by elevated levels of pre-weaning mortality.

## Figures and Tables

**Figure 1 animals-12-00847-f001:**
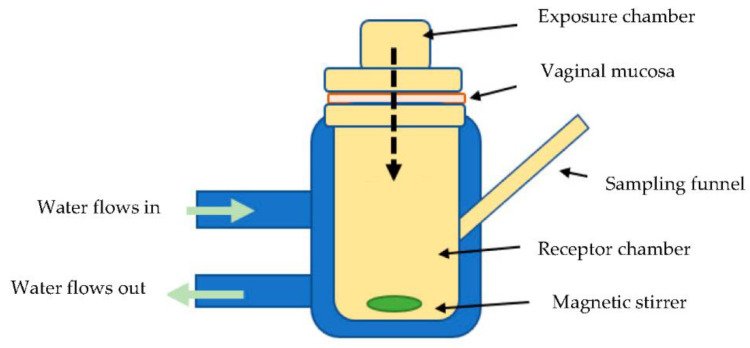
Schematic diagram of Franz diffusion cell.

**Figure 2 animals-12-00847-f002:**
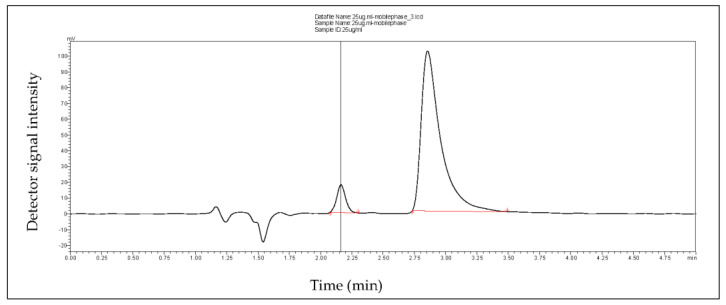
Chromatogram of dexamethasone treatment.

**Figure 3 animals-12-00847-f003:**
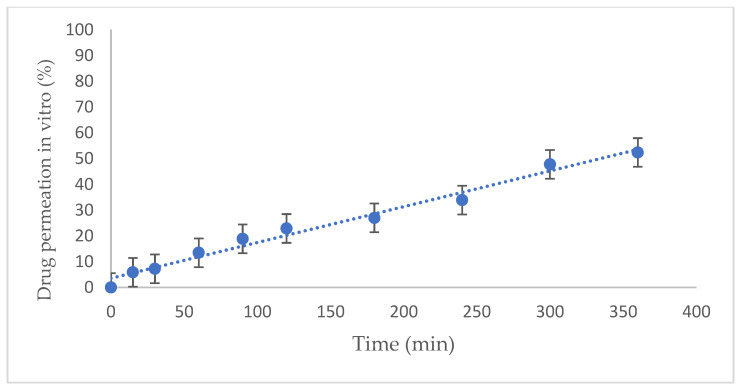
In vitro permeation of dexamethasone treatment (%) through porcine vaginal mucosa using Franz diffusion cells over time (minutes). Simulated vaginal fluid was used for the donor and acceptor solutions, and cells were incubated at 37 °C ± 1 °C. The diffusion tests were run six times and are presented as the mean ± standard deviation (SD) of the mean.

**Figure 4 animals-12-00847-f004:**
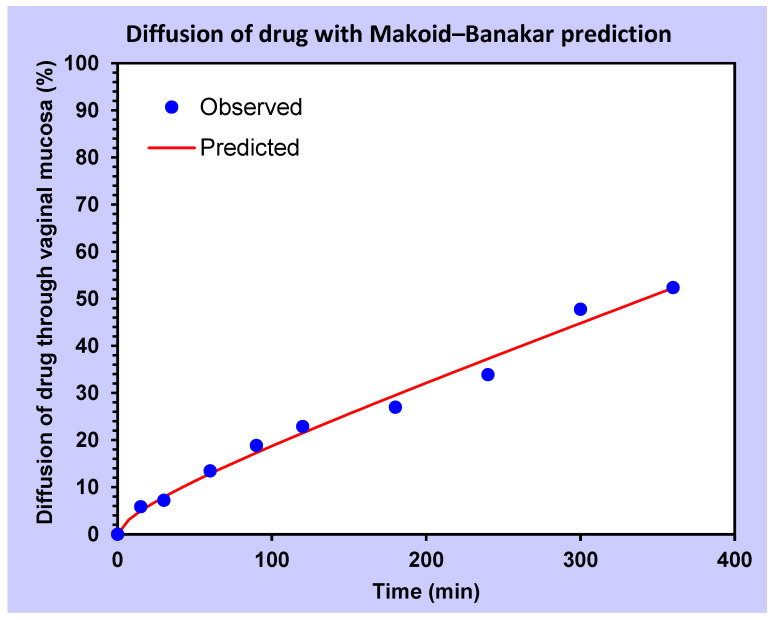
A diffusion profile of dexamethasone treatment through porcine vaginal mucosa in-vitro fitted by Makoid–Banakar model (*F* = *kMB* × *t**^n^* × Exp(−*k* × *t*)). *r*^2^ adjusted = 0.9871 and AIC = 39.2.

**Table 1 animals-12-00847-t001:** Ethogram used for monitoring farrowing behaviour over a 24 h period.

	Behaviour	Description
Posture	Stand	Sow is standing.
	Sit	Sow is sitting.
	Side lie	Lateral recumbency: udder or at least the top line or teats are not obscured.
	Belly lie	Sternal recumbency: the udder is obscured under the sow.
Spontaneous behaviours		
	Tail flick	The tail is moved rapidly up and down.
	Back leg forward	In a lateral lying position, the back leg is pulled forwards and/or in towards the body.
	Back arch	In a lateral lying position, one or both sets of legs become tense and are pushed away from the body and in towards the center, forming an arch in the back.
	Paw	The sow uses the forepaw to scrape the floor in a pawing position.
	Piglet-directed aggression	The sow snout flicks quickly behind/snaps at the approaching piglet.
	Overlay	Any event where a piglet is being crushed by the sow. Piglets may be under the sow, squashed at the front of the crate or under the trotter.

**Table 2 animals-12-00847-t002:** Effects of dexamethasone administered the day of farrowing (0700 h, gestation day 115) as a vulval injection (DexInj), applied topically into the vagina (DexTop), or no treatment (Control) on mean (±SE) sow and piglet performance indicators.

	Control	DexInj	DexTop	*p* Value
Farrowing duration (min)	232 ± 26	157 ± 42	170 ± 31	0.214
Piglet birth interval (min)	17.1 ± 1.8	13.2 ± 1.7	14.1 ± 1.7	0.289
Incidence of dystocia (%)	47 ± 11	21 ± 8	42 ± 11	0.263
Incidence of stillbirth (%)	67 ± 10	68 ± 9	57 ± 11	0.655
Total piglets born	11.7 ± 0.5	11.7 ± 0.5	11.8 ± 0.3	0.943
Total piglets born alive	11.3 ± 0.6	11.1 ± 0.6	11.0 ± 0.6	0.952
Colostrum intake (g)	319.3 ± 6.7	313.5 ± 6.5	320.3 ± 6.5	0.718
Incidence of overlay in 24 h (%)	57 ± 11	39 ± 10	57 ± 11	0.393
Piglet survival to 24 h (%)	89.6 ± 2.8	91.5 ± 2.7	90.9 ± 2.8	0.872
Litter size weaned	10.4 ± 0.3	10.6 ± 0.3	10.6 ± 0.3	0.855
Survival of piglets to weaning (%)	79.9 ± 2.9	88.8 ± 2.9	86.6 ± 3.0	0.094

**Table 3 animals-12-00847-t003:** Effects of dexamethasone administered the day of farrowing (0700 h, gestation day 115) as a vulval injection (DexInj), applied topically into the vagina (DexTop), or no steroid treatment (Control) on mean (±SE) sow behaviours during 24 h from onset of farrowing. The number of sows that displayed any piglet-directed aggression is expressed as a percentage over the total treatment group (95% CI). In addition, the time each sow spent in the laying position (udder exposed) is presented as a percentage over the total 24 h observation period.

	Control	DexInj	DexTop	*p* Value
Back arch	9.9 ± 3.7	4.1 ± 1.6	5.8 ± 2.4	0.256
Leg up	12.3 ± 4.9	4.3 ± 1.8	7.6 ± 2.9	0.199
Pawing	8.8 ± 3.2	6.0 ± 2.5	3.4 ± 1.5	0.321
Tail flick	0.1 ± 0.1	1.2 ± 0.5	0.4 ± 0.1	0.088
Total pain behaviours	22.5 ± 4.6	15.5 ± 4.9	19.0 ± 4.6	0.525
Total position changes	77.3 ± 11.8	53.8 ± 11.8	69.3 ± 12.3	0.382
Piglet-directed aggression (%)	50 ± 5	25 ± 4	22 ± 4	0.269
Time spent on side (%)	81.6 ± 3.6	87.8 ± 3.6	82.8 ± 3.8	0.350

## Data Availability

The data presented in this study are available on request from the corresponding author.

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
