# Peer review of "Effect of Dexamethasone and Route of Administration on Sow Farrowing Behaviours, Piglet Delivery and Litter Performance"

_animals, 2022, doi:10.3390/ani12070847_

Round 1

Reviewer 1 Report

Comments to the Authors of manuscript number: animals-1606763 entitled “Effect of dexamethasone and route of administration on sow farrowing behaviors, piglet delivery and litter performance.”.

Authors have presented the study performed on sows (gilts and after first parity) who suffer from pain during parturition. They hypothesized that dexamethasone given during parturition could decrease the pain and influence behaviour. It is very surprising but very interesting study. Until, dexamethasone was used (by Caroll, who e.g. gave it within the first hour to newborns) to investigate the performance or prenatal programming in swine model.

I have one doubt. Why Authors presented the preparation using the producer`s name, but They did not write that it is dexamethasone. Is it a advertisement? It should be corrected along the text. Moreover, strictly should be written what form it is of dexamethasone.

  1. L 58-59- if Authors wrote “previous investigations” they should be cited in this line.
  2. L 63- “its biological half-life” does it relate to dexamethasone? It should be written clearly.
  3. L 100 – what is receptor fluid. It is not clear.
  4. L 136 – all data about Dexapent TM should be given: producer and the composition.
  5. L L 224 – to better comparison not only chemical composition of preparation by Zang should be given. Readers do not know what is DexapentTM

Author Response

Thankyou for your comments, please see answers to critiques in the attached file. 

Reviewer 2 Report

Ward et al.

Animals-1606763

Effect of Dexamethasone and Route of Administration on Sow Farrowing Behaviors, Piglet Delivery, and Litter Performance. 

Major (General or Overall) Comments to the Authors

The authors set out to evaluate the effect of intravaginal and intramuscular Dexamethasone on sow behavior and piglet neonatal survival. The manuscript is overall well written, and the authors did a good job explaining the relevance of this topic. Hypothesis and objectives were clearly stated, and overall, the manuscript is in good shape.

Specific comments by section/line.

Introduction

In general, the introduction is easy to follow and well organized.

Materials and Methods

Overall, the materials and methods used in the study were well described. My only concern is with the experimental design. The sample size of this study seems small, particularly for binary outcomes. This is the reason why it is important for the authors to clarify which was the primary outcome, and how was the sample size calculation performed to evaluate such outcome (see below). Also, why didn’t the authors measure Dexamethasone in blood? That way you could have better evaluated vaginal absorption of the drug (in vivo evaluation to complement in vitro evaluation). Please clarify.

Ln 125 Missing a parenthesis after “mg”.

Ln 159 The authors mention that a subset of sows (n = 26) was video recorded, but do not specify how many for each treatment group. Please clarify.

Ln 165-175 The statistics section needs some more work. First, which was the primary outcome? Which were the secondary outcomes? Did the authors perform a sample size calculation? If so, how was it performed? How were animals randomized into treatments? How were binary outcomes analyzed? Are results presented as LSM or arithmetic means? In addition, please add in between parentheses which outcomes you are referring to for each analysis, e.g., “measurements were analyzed using a linear mixed model”, which measurements? Same for “piglet data” and “behavioral count data”.

Table 2 Please clarify how is this information relevant to this study. I consider this table should be added as a supplementary table.

Results

The results are overall well summarized and described.

Table 4 Please add units to each item as done in Table 3.

Discussion/Conclusion

Overall, the discussion and conclusion sections are well written. Main points were correctly addressed and discussed. The authors also did a good job comparing their observations to that reported in the existing literature. My only question to the authors is: could the lack of significance in some of the outcomes be attributed to the small sample size of the study?  If so, please discuss in this section.

Author Response

Thankyou for your edits, please see answers to critiques in attached file. 

Round 2

Reviewer 2 Report

All of my comments have been adequately addressed by the authors, and the manuscript is now in great shape. Some minor edits are required in the stats section:

Ln 171-172; what do you mean by "Other outcomes of treatment"? Did you mean "Other outcomes of interest"? Please revise.

Ln 174 "All sow data was fit random terms sow identification and room.." Did you mean that sow ID and room were included in the model as random effects? If so, please revise this sentence to improve clarity.  

Author Response

Thank you again for your time in reviewing the paper, please see responses to attached comments: 

Ln 171-172; what do you mean by "Other outcomes of treatment"? Did you mean "Other outcomes of interest"? Please revise.

Yes, has been amended to "other outcomes of interest' 

Ln 174 "All sow data was fit random terms sow identification and room.." Did you mean that sow ID and room were included in the model as random effects? If so, please revise this sentence to improve clarity.  

"Yes that is correct, this sentence has now been modified to make it clearer to the reader'